# Semantic Probabilistic Layers
# for Neuro-Symbolic Learning

**Kareem Ahmed**
CS Department
UCLA
ahmedk@cs.ucla.edu

**Stefano Teso**
CIMeC and DISI
University of Trento
stefano.teso@unitn.it

**Kai-Wei Chang**
CS Department
UCLA
kwchang@cs.ucla.edu

**Guy Van den Broeck**
CS Department
UCLA
guyvdb@cs.ucla.edu

**Antonio Vergari**
School of Informatics
University of Edinburgh
avergari@ed.ac.uk

## Abstract

We design a predictive layer for structured-output prediction (SOP) that can be plugged into any neural network guaranteeing its predictions are consistent with a set of predefined symbolic constraints. Our **S**emantic **P**robabilistic **L**ayer (SPL) can model intricate correlations, and hard constraints, over a structured output space all while being amenable to end-to-end learning via maximum likelihood. SPLs combine exact probabilistic inference with logical reasoning in a clean and modular way, learning complex distributions and restricting their support to solutions of the constraint. As such, they can faithfully, and efficiently, model complex SOP tasks beyond the reach of alternative neuro-symbolic approaches. We empirically demonstrate that SPLs outperform these competitors in terms of accuracy on challenging SOP tasks including hierarchical multi-label classification, pathfinding and preference learning, while retaining *perfect* constraint satisfaction. Our code is made publicly available on Github at github.com/KareemYousrii/SPL.

## 1 Introduction

Modularity is among the major factors that propelled the Cambrian explosion of deep learning [35]. By stacking multiple *differentiable* layers together, practitioners are able to train deep classifiers in an end-to-end fashion with little effort. However, despite its flexibility, *this modular approach to learning does not guarantee that the predictions of these models conform to our expectations of what makes sense*. On the contrary, unconstrained deep classifiers are notorious for leading to predictions that are inconsistent with the logical constraints governing an underlying domain.

This is even more evident in, and crucial for, structured output prediction (SOP) tasks, where classifiers have to predict hundreds of mutually constrained labels [73, 8]. Consider for example a classical SOP task such as multi-label classification (MLC) [74]. Learning a multi-label classifier that disregards the correlations among labels, e.g., by considering them *fully independent* given the inputs, yields sub-optimal results [6]. In more challenging tasks such as hierarchical MLC (HMLC) [71] or pathfinding [60], leveraging the domain's logical constraints (encoding, e.g., the label hierarchy or acyclicity and connectedness of a path) at training time can improve prediction accuracy [44], but it cannot guarantee that the predictions are always *consistent* with the constraints at inference time [32]. Fig. 1 illustrates this problem in the context of pathfinding: constraint-unaware neural networks systematically fail to predict label configurations that form a valid path. In many safety-critical

36th Conference on Neural Information Processing Systems (NeurIPS 2022).



| GROUND TRUTH | RESNET-18 | SEMANTIC LOSS | SPL (ours) |

Figure 1: **Neural nets struggle with satisfying validity constraints in complex semantic SOP tasks** such as predicting the lowest-cost path from the top-left to the bottom-right corners of a Warcraft map. Even state-of-the-art neuro-symbolic approaches like the Semantic Loss [80] fail to ensure consistency with hard rules (c). SPLs in contrast guarantees validity while retaining modularity, expressiveness and efficiency. See Sec. 5 for complete experimental details and additional results.

scenarios such as protein function [63] and interaction prediction [65], and drug discovery [20, 24], predicting inconsistent solutions can not only be harmful but also highly expensive [5, 34].

Unsurprisingly, due to their discrete nature, injecting logical constraints into deep neural networks while retaining modularity and differentiability is extremely challenging, as demonstrated in the *neuro-symbolic learning* literature [66]. One such attempt has been to learn neural networks that satisfy the logical constraints by explicitly minimizing a differentiable loss term encoding the probability that the networks violates the constraint for a given prediction. And while successful, such approaches do not guarantee consistency of the predictions at test time. More recently, researchers have proposed predictive layers that do guarantee consistency, but these are restricted to specific kinds of symbolic knowledge [32, 70] or become intractable for even moderately complex logical constraints [38].

Motivated by these observations, we introduce a novel **S**emantic **P**robabilistic **L**ayer (SPL) for modeling intricate correlations, and logical constraints on the labels of the output space in a modular and probabilistically sound manner. It does so by leveraging recent advancements in the literature on probabilistic circuits [76, 13]. The key features of SPL are that, on the one hand, it can be used as a *drop-in replacement* for common predictive layers of deep nets like sigmoid layers, and on the other, it *guarantees* the output's consistency with any prespecified logical constraints. Importantly, SPL also supports efficient inference and – perhaps surprisingly – does not complicate training.

**Contributions.** Summarizing, we: (*i*) Identify six desiderata that neuro-symbolic predictors ought to satisfy to flexibly and reliably support real-world SOP tasks, and show that state-of-the-art approaches fall short of one or more of them (Table 1); (*ii*) Introduce SPL, a novel semantic probabilistic layer that leverages probabilistic circuits to satisfy all six desiderata, i.e., that can be plugged into neural networks to ensure predictions to comply to given logical constraints, while retaining efficiency, expressivity, differentiability, and fully probabilistic semantics; (*iii*) We provide empirical evidence of the effectiveness of SPL in several challenging neuro-symbolic SOP tasks, such as HMLC and pathfinding, where it outperforms state-of-the-art neuro-symbolic approaches, often by a noticeable margin. We implemented SPLs in PyTorch [54], and our code is made publicly available on Github at github.com/KareemYousrii/SPL.

## 2 Designing a probabilistic layer for neuro-symbolic SOP

**Notation.** In the following, we denote scalar constants $x$ in lower case, random variables $X$ in upper case, vectors of constants $\boldsymbol{x}$ in bold and vectors of random variables $\mathbf{X}$ in capital boldface. $\mathbb{1}\{\varphi\}$ denotes the indicator function that evaluates to 1 if the statement $\varphi$ holds and to 0 otherwise. We denote by $\boldsymbol{x} \models \mathsf{K}$ that the value assignment $\boldsymbol{x}$ to variables $\mathbf{X}$ satisfies a logical formula $\mathsf{K}$.

**Neuro-symbolic SOP.** We tackle SOP tasks in which a neural net classifier must learn to associate instances $\boldsymbol{x} \in \mathbb{R}^D$ to $L$ *interdependent* labels, identified by the vector $\boldsymbol{y} \in \{0, 1\}^L$. We assume that we can abstract any neural classifier into two components: a feature extractor $f$ that maps inputs $\mathbf{X}$ to a $M$-dimensional embedding $\mathbf{Z} = f(\mathbf{X})$ and a predictive final layer that outputs the label distribution $p(\mathbf{Y} \mid \mathbf{Z})$. For example, the simplest, and yet widely adopted [51, 80, 32], predictive layer in neural classifiers for SOP considers labels $Y_i$ to be conditionally independent from each other given $\mathbf{Z}$, i.e., $p(\mathbf{Y} \mid \mathbf{Z}) = \prod_{i=1}^{L} p(Y_i \mid \mathbf{Z})$. We refer to this as *fully independent layer* (FIL). In a FIL,

Table 1: **SPL is the only approach to satisfy all the desiderata for neuro-symbolic SOP.** An in-depth discussion of all competitors can be found in Sec. 4.

| DESIDERATUM | LOSSES | | | LAYERS | | | | |
|---|---|---|---|---|---|---|---|---|
| | DL2 [29] | SL [80] | NeSyEnt [3] | FIL | EBM [43] | MultiplexNet [38] | CCN [33] | SPL (ours) |
| (D1) Probabilistic | ✗ | ✓ | ✓ | ✓ | ✗ | ✓ | ✗ | ✓ |
| (D2) Expressive | ✗ | ✗ | ✗ | ✗ | ✓ | ✗ | ✗ | ✓ |
| (D3) Consistent | ✗ | ✗ | ✗ | ✗ | ✗ | ✓ | ✓ | ✓ |
| (D4) General | ✓ | ✓ | ✓ | ✗ | ✓ | ✓ | ✗ | ✓ |
| (D5) Modular | ✓ | ✓ | ✓ | ✓ | ✓ | ✓ | ✓ | ✓ |
| (D6) Efficient | ✓ | ✓ | ✓ | ✓ | ✗ | ✗ | ✓ | ✓ |

$p(Y_i = y_i \mid \boldsymbol{z})$ is computed as $\sigma(\mathbf{w}_i^\top \boldsymbol{z})$ where $\mathbf{w}_i \in \mathbb{R}^M$ is a vector of parameters and $\sigma(x)$ is the logistic sigmoid function $1/(1 + e^{-x})$.

We are interested in dependencies between labels that can occur both as *correlations*, as is the case in MLC [22], and as *logical constraints* encoded by logical formulas. For example, in a HMLC task [32] one logical constraint can encode the fact that observing a label for the class cat and dog, implies observing the label for their superclass animal

$$(Y_{\text{cat}} = 1 \implies Y_{\text{animal}} = 1) \wedge (Y_{\text{dog}} = 1 \implies Y_{\text{animal}} = 1). \tag{1}$$

Specifically, we assume symbolic knowledge to be supplied in the form of constraints encoded as a logical formula denoted as $\mathsf{K}$ and defined over the labels $\mathbf{Y}$ and optionally over a subset of the discrete input variables in $\mathbf{X}$, if any (e.g., in our experiments, the predicted simple path is constrained to lie within the subset of edges appearing in the input graph, see Sec. 5). On the other hand, we expect a model to learn the label correlations from data. We call such task *neuro-symbolic SOP*.

**Desiderata for neuro-symbolic SOP.** To tackle this setting, we seek an algorithmic strategy for replacing the predictive layer in any neural network classifier with little effort, with the aim of injecting complex symbolic knowledge and allowing for flexible probabilistic reasoning. We formalize these observations into the following six desiderata for our predictive layer:

- D1. **Probabilistic**: The layer should enjoy sound probabilistic semantics, and deliver normalized probabilistic predictions to facilitate maximum-likelihood learning and sound decision making by virtue of calibrated probabilistic predictions

- D2. **Expressive**: It should be able to compactly encode intricate *correlations* between labels.

- D3. **Consistent**: It should always output predictions that are consistent with the prespecified symbolic knowledge, i.e., for all $\boldsymbol{x}$ and $\boldsymbol{y}$, if $(\boldsymbol{x}, \boldsymbol{y}) \not\models \mathsf{K}$ then $p(\boldsymbol{y} \mid \boldsymbol{x}) = 0$.

- D4. **General**: It should support rich *logical constraints* over the labels expressed in some formal language, e.g., propositional logic.

- D5. **Modular**: It should be applicable to any off-the-shelf (and possibly pretrained) neural network in a modular fashion, enabling end-to-end learning and rapid prototyping.

- D6. **Efficient**: The time required by the predictor to compute a prediction should be linear in the size of the predictor and of the hard constraint representation.

For example, FILs are clearly probabilistic (D1), modular (D5), and efficient (D6), but at the cost of being incapable of modeling intricate correlations and logical constraints and thus generating inconsistent predictions (D2–D4) (see also Fig. 1). Table 1 summarizes how the other popular and effective approaches to neuro-symbolic SOP nowadays fall short of one of more desiderata as well. We discuss this in detail in Sec. 4. To the best of our knowledge, our proposed *semantic probabilistic layers* (SPLs) are the first algorithmic solution to satisfy all above desiderata.

**SPL.** At a high level, SPL realizes the above desiderata in a single layer that combines exact probabilistic inference with logical reasoning in a clean and modular way, learning complex distributions and restricting their support to solutions of the constraint.

**Definition 2.1** (Semantic probabilistic layer (SPL)). Given an input configuration $\boldsymbol{x}$, a SPL decomposes the computation of the probability of a label configuration as:

$$p(\boldsymbol{y} \mid f(\boldsymbol{x})) = q_{\Theta}(\boldsymbol{y} \mid f(\boldsymbol{x})) \cdot c_{\mathsf{K}}(\boldsymbol{x}, \boldsymbol{y}) / \mathcal{Z}(\boldsymbol{x}) \quad \text{where} \quad \mathcal{Z}(\boldsymbol{x}) = \sum_{\boldsymbol{y}} q_{\Theta}(\boldsymbol{y} \mid \boldsymbol{x}) \cdot c_{\mathsf{K}}(\boldsymbol{x}, \boldsymbol{y}). \tag{2}$$

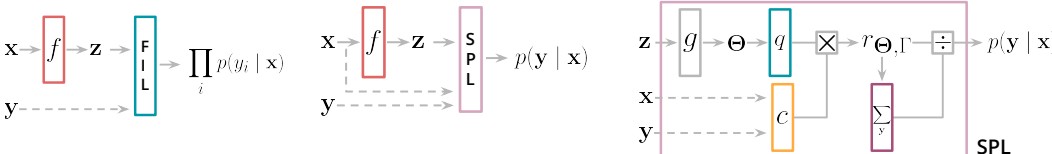

Figure 2: **A high level view of SPLs.** The predictive layer of a neural network for neuro-symbolic SOP, e.g., a FIL (**left**), can be readily replaced by a SPL (**middle**). SPLs are implemented (**right**) by multiplying together a probabilistic circuit $q_\Theta(\mathbf{Y} \mid f(\mathbf{X}))$ parameterized by (a function $g$ of) the network's embeddings $f(\mathbf{X})$, and a constraint circuit $c_\mathsf{K}(\mathbf{X}, \mathbf{Y})$ embodying the symbolic knowledge. The result is normalized by efficiently marginalizing over the product circuit $r_{\Theta,\mathsf{K}}$, so as to guarantee fully probabilistic semantics and end-to-end differentiable learning by maximum likelihood.

Here, $q_\Theta(\boldsymbol{y} \mid f(\boldsymbol{x}))$ is a module to perform probabilistic reasoning by encoding an expressive distribution over the labels parameterized by $\Theta$; $c_\mathsf{K}(\boldsymbol{x}, \boldsymbol{y})$ is a module to ensure consistency of the predictions by encoding logical constraints $\mathsf{K}$ and being non-zero only when $\mathsf{K}$ is satisfied, i.e., $c_\mathsf{K}(\boldsymbol{x}, \boldsymbol{y}) = \mathbb{1}\{(\boldsymbol{x}, \boldsymbol{y}) \models \mathsf{K}\}$; and $\mathcal{Z}(\boldsymbol{x})$ is a renormalization term, also called the partition function. It is worth noting that this amounts to taking a product of experts [37] which is, in general, hard.

Fig. 2 illustrates the computational graph of our SPL at training time. In order to satisfy all D1-D6, we will realize both $q_\Theta$ and $c_\mathsf{K}$ as *circuits* [76, 13], constrained computational graphs that enable tractable computations. Differently from FILs, $q_\Theta$ in SPLs can encode an expressive joint distributions over the labels and therefore attain full expressiveness by scaling the number of parameters $\Theta$ (D2). Consistency is guaranteed by the component $c_\mathsf{K}$: by multiplying it to the joint probability of a label configuration the resulting product distribution $r_{\Theta,\mathsf{K}}(\boldsymbol{y}, \boldsymbol{x}) = q_\Theta(\boldsymbol{y} \mid f(\boldsymbol{x})) \cdot c_\mathsf{K}(\boldsymbol{x}, \boldsymbol{y})$ will have its support effectively cut by $\mathsf{K}$, and thus cannot allocate any probability mass to inconsistent predictions (D3). Additionally, $c_\mathsf{K}$ will allow to encode general propositional logical constraints in a compact computational graph (D4). Lastly, the product $r_{\Theta,\mathsf{K}}(\boldsymbol{x}, \boldsymbol{y})$ is fully differentiable and allows SPL to be an off-the-shelf replacement for other predictive layers (see Fig. 2) and enables end-to-end learning (D5). By renormalizing $r_{\Theta,\mathsf{K}}(\boldsymbol{x}, \boldsymbol{y})$ and outputting normalized probabilities, SPL enables the exact computation of gradients for $\Theta$, which can therefore be trained by maximum likelihood (D1).

Thanks to recent advancements in the literature on circuits, we can compute the partition function $\mathcal{Z}(\boldsymbol{x})$ efficiently in time linear in the size of $r_{\Theta,\mathsf{K}}$, thus preserving efficiency (D6) and not compromising on the other desiderata. This will also yield correct (and consistent) predictions at test time, when an SPL computes the MAP state $\boldsymbol{y}^* = \mathrm{argmax}_{\boldsymbol{y}} \, r_{\Theta,\mathsf{K}}(\boldsymbol{y}, \boldsymbol{x}) / \sum_{\boldsymbol{y}} r_{\Theta,\mathsf{K}}(\boldsymbol{y}, \boldsymbol{x})$. The next section clarifies *how* to implement the modules of SPL as circuits while satisfying these desiderata.

## 3  Realizing SPLs with tractable circuit representations

The components of SPLs are *circuits*, a large class of computational graphs that can represent both functions and distributions [13, 18]. Circuits subsume many tractable generative and discriminative probabilistic models—from Chow-Liu and latent tree models [14, 11], to hidden Markov models (HMMs) [62], sum-product networks (SPNs) [61], decision trees [41, 15], and deep regressors [40]—as well as many compact representations of logical formulas, such as (ordered) binary decision diagrams [4], sentential decision diagrams (SDDs) [17] and others [18].

The key idea behind SPLs is to leverage this single formalism to represent both an expressive joint distribution for $q_\Theta(\boldsymbol{y} \mid f(\boldsymbol{x}))$ and a compact encoding of the logical constraints for $c_\mathsf{K}(\boldsymbol{x}, \boldsymbol{y})$, while ensuring the exact and efficient evaluation of Eq. (2). This can be achieved by ensuring that these computational graphs abide certain structural properties: *smoothness*, *decomposability*, *determinism* and *compatibility* [18, 75]. Next, we introduce *probabilistic circuits* for modeling $q_\Theta$ (Sec. 3.1) and *constraint circuits* for $c_\mathsf{K}$ (Sec. 3.2), while in Sec. 3.4 we propose a more efficient implementation of SPL utilizing a single circuit.

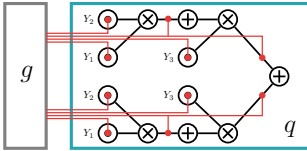 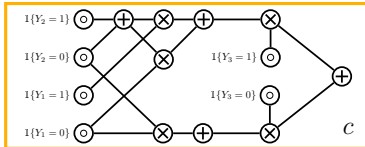

Figure 3: **Examples of circuits in SPL**. **Left**: a neural conditional probabilistic circuit $q_\Theta$. Red lines indicate how the output of $g$ parameterizes the input distribution parameters $\boldsymbol{\lambda}$ and the sum unit parameters $\boldsymbol{\omega}$ of $q$, both indicated as red dots. **Right**: constraint circuit encoding the logical constraint of Eq. (1) where labels are $Y_i \in \{Y_{\text{cat}}, Y_{\text{dog}}, Y_{\text{animal}}\}$. Note that $q$ and $c$ are smooth, decomposable (Def. 3.3) and compatible (Def. 3.7) and $c$ is deterministic (Def. 3.6). By parameterizing $c$ via $g$ we can obtain a single-circuit SPL (Sec. 3.4). **Both**: circuits $q$ and $c$ are compatible, as product units with the same scope decompose in the same way. E.g., consider the first two product units of $q$ and $c$, right to left and top to bottom. Both units decompose $\{Y_3, Y_2, Y_1\}$ into $Y_3$ and $Y_2, Y_1$.

## 3.1 Representing expressive distributions with probabilistic circuits

We start by introducing circuits for *joint* probability distributions, and then extend the discussion to *conditional* distributions, which we use to implement $q_\Theta(\mathbf{Y} \mid f(\mathbf{X}))$ in SPLs.

**Definition 3.1** (Circuits). A circuit $h$ over variables $\mathbf{Y}$ is a computational graph encoding a parameterized function $h_\Theta(\mathbf{Y})$ by combining three kinds of computational units: input functional units, sum units, and product units. An input functional $n$ represents a base parametric function $h_n(\text{sc}(n); \boldsymbol{\lambda})$ over some variables $\text{sc}(n) \subseteq \mathbf{Y}$, called its scope, and it is parameterized by $\boldsymbol{\lambda}$. Sum and product units $n$ elaborate the output of other units, denoted $\text{in}(n)$. Sum units are parameterized by $\boldsymbol{\omega}$ and compute the weighted sum of their inputs $\sum_{c \in \text{in}(n)} \omega_c h_c(\text{sc}(n))$, while product units compute $\prod_{c \in \text{in}(n)} h_c(\text{sc}(n))$. The parameters $\Theta$ of a circuit encompass the parameters of all input functionals ($\boldsymbol{\lambda}$) and the parameters of sum units ($\boldsymbol{\omega}$).

For any input $\boldsymbol{y}$, the value of $h_\Theta(\boldsymbol{y})$ can be evaluated by propagating the output of the input units through the computational graph and reading out the value of the last unit. The *support* of $h$ is the set of all states $\boldsymbol{y}$ of $\mathbf{Y}$ for which the output is non-zero, i.e., $\text{supp}(h) = \{\boldsymbol{y} \mid h(\boldsymbol{y}) \neq 0\}$.

**Definition 3.2** (Probabilistic circuits (PCs)). A circuit $q$ is a PC if it encodes a (possibly unnormalized) probability distribution, i.e., $q_\Theta(\boldsymbol{y})$ is non-negative for all configurations $\boldsymbol{y}$ of $\mathbf{Y}$.

From here on, we will assume PCs to have positive sum parameters $\boldsymbol{\omega}$ and whose input units model valid distributions, e.g., Bernoullis, as these conditions are sufficient for satisfying Def. 3.2. Moreover, w.l.o.g. we will assume the sum and product units to be organized into alternating layers, and that every product unit $n$ receives only two inputs $c_1, c_2$, i.e., $q_n(\mathbf{X}) = q_{c_1}(\mathbf{Y}) \cdot q_{c_2}(\mathbf{Y})$. These conditions can easily be enforced in polynomial time [77, 78]. We are specifically interested in smooth and decomposable PCs, as they will be enabling efficient inference in SPL (Sec. 3.3).

**Definition 3.3** (Smoothness & Decomposability). A circuit is *smooth* if for every sum unit $n$, its inputs depend on the same variables: $\forall c_1, c_2 \in \text{in}(n), \text{sc}(c_1) = \text{sc}(c_2)$. It is *decomposable* if the inputs of every product unit $n$ depend on disjoint sets of variables: $\text{in}(n) = \{c_1, c_2\}, \text{sc}(c_1) \cap \text{sc}(c_2) = \emptyset$.

Smooth and decomposable PCs are both expressive and efficient: they can encode distributions with hundred millions of parameters and be effectively learned by gradient ascent [58]. The structure of their computational graph can be either specified manually [61, 58, 56] or acquired automatically from data [77, 64, 16], e.g., by first learning a latent tree model and then compiling the latter into a circuit [46]. These circuits are competitive with intractable models such as variational autoencoders and normalizing flows scores on several benchmarks [45].

As proposed by Shao et al. [68], any (smooth and decomposable) PC $q_\Theta(\mathbf{Y})$ encoding a *joint* distribution over the labels $\mathbf{Y}$ can be turned into a (smooth and decomposable) *conditional* circuit, conditioned by input variables $\mathbf{X}$, by letting its parameters be a function of $\mathbf{X}$.

**Definition 3.4** (Neural conditional circuits [67]). A conditional circuit $q(\mathbf{Y}; \Theta = g(\mathbf{X}))$ models the conditional distribution $p(\mathbf{Y} \mid \mathbf{X})$ via a differentiable function $g$ that maps every input configuration $\boldsymbol{x}$ to the set of parameters of $\Theta$ of $p$, also called the *gating function*.

An example of a smooth and decomposable conditional circuit is shown in Fig. 3. This design immediately allows us to implement $q_{\Theta}(\mathbf{Y} \mid f(\mathbf{X}))$ in SPL as a conditional PC whose gating function maps the feature embedding space $\mathbb{R}^K$ to the parameter space $\mathbb{R}_+^{|\Theta|}$, realizing $q(\mathbf{Y}; \Theta = g(f(\mathbf{X})))$. As such, the gating function $g$ creates a clean interface between any pre-trained feature extractor $f$ and the PC $q$ (Fig. 3). While one can devise $g$ in several ways, we strive for simplicity in our experiments and adopt vanilla multi-layer perceptrons (MLPs) whose final activations are either sigmoids, if they have to predict the parameters $\boldsymbol{\lambda}$ of the Bernoulli input distributions of $q$, or softmax, if they output the sum unit parameters $\boldsymbol{\omega}$ (Def. 3.1).

## 3.2 Encoding logical formulas with constraint circuits

The next step is to translate a logical constraint K into a smooth and decomposable circuit $c_{\mathsf{K}}(\boldsymbol{x}, \boldsymbol{y})$. To this end, we employ a special type of PCs, defined as follows.

**Definition 3.5** (Constraint circuits). A PC $c$ over variables $\mathbf{X} \cup \mathbf{Y}$ is a constraint circuit encoding prior knowledge K if it computes $\mathbb{1}\{(\boldsymbol{x}, \boldsymbol{y}) \models \mathsf{K}\}$ for every configuration $(\boldsymbol{x}, \boldsymbol{y})$.

As a practical way to realize such a circuit, we will consider constraint circuits that have all sum unit parameters equal to 1 and input functionals that are indicator functions over their scope, e.g., $c_n(\boldsymbol{z}) = \mathbb{1}\{\boldsymbol{z} \models \varphi(n)\}$ where $\mathbf{Z}$ is the scope of the input and $\varphi(n)$ a constraint over it. Furthermore, we require each sum unit in it to be *deterministic*.

**Definition 3.6** (Determinism). A sum unit $n$ is *deterministic* if its inputs have disjoint supports, i.e., $\forall c_1, c_2 \in \mathsf{in}(n), c_1 \neq c_2 \implies \mathsf{supp}(c_1) \cap \mathsf{supp}(c_2) = \emptyset$.

Fig. 3 shows an example of a deterministic constraint circuit. Thanks to determinism, we can readily translate classical compact representations for logical formulas such as (ordered) binary decision diagrams [4, 9] and sentential decision diagrams (SDDs) [17] into constraint circuits as defined above. This becomes evident when they are written in the language of negation normal form [18] and their *and* gates (resp. *or* gates) are replaced with product units (resp. sum units) [13]. A logic constraint can therefore be represented as a constraint circuit for SPLs, by utilizing any of the many tools available for OBDDs [72] or SDDs [10, 53]. Sec. B illustrates in detail how to compile the example constraint of Eq. (1) into the constraint circuit of Fig. 3 in this way.

The worst-case size of the constraint circuit depends on a) the algorithm employed for compilation and, b) the local structure of the constraints, rather than the number of labels. For example, in our Warcraft experiment (see Sec. 5), we have a label configuration space over edges in a $12 \times 12$ grid, yielding $2^{12^2} = 2^{144} \approx 10^{43}$ states. However, only $10^{10}$ configurations satisfy the constraint that these edge labels form a valid path in the grid. If our compilation algorithm were to simply enumerate these configurations, putting them in a logical OR (as done in some neuro-symbolic learners such as MultiplexNet [38]), the size of the constraint circuit, denoted as $|c|$, would be $10^{10}$. However, by using recent advancements in compiling logical formulas into constraints circuits, we can can greatly reduce the circuit size. For example, compiler we use [1] generates circuits whose size is worst-case exponential in the treewidth of the CNF representation of the logical formula, but typically much smaller. See Sec. 5 and Sec. F for details.

## 3.3 Efficient inference in SPLs

As discussed above, PCs can be expressive (D2) and are modular (D5), while constraint circuits ensure consistency (D3) for general constraints (D4). What remains to be shown to complete SPLs is that the product supports efficient normalization (D1) and inference (D6), specifically that it allows for the efficient evaluation of the normalization constant of $r_{\Theta,\mathsf{K}}$, and its MAP state. To this end, we need to introduce the notion of compatibility between the two circuits [75].

**Definition 3.7** (Compatible circuits in SPLs). A smooth and decomposable conditional PC $q(\mathbf{Y}; \Theta)$ is compatible over variables $\mathbf{Y}$ with a smooth and decomposable constraint circuit $c_{\mathsf{K}}(\mathbf{Y}, \mathbf{X})$ if any pair of product units $n \in q$ and $m \in c_{\mathsf{K}}$ with the same scope over $\mathbf{Y}$ can be rearranged to be mutually compatible and decompose in the same way: $(\mathsf{sc}(n) = \mathsf{sc}(m)) \implies (\mathsf{sc}(n_i) = \mathsf{sc}(m_i), n_i$ and $m_i$ are compatible) for some rearrangement of the inputs of $n$ (resp. $m$) into $n_1, n_2$ (resp. $m_1, m_2$). The two circuits $q$ and $c$ shown in Fig. 3 are compatible.

Table 2: SPLs outperform all loss-based competitors in the neuro-symbolic benchmarks of [80].

| | SIMPLE PATH | | | PREFERENCE LEARNING | | |
|---|---|---|---|---|---|---|
| ARCHITECTURE | EXACT | HAMMING | CONSISTENT | EXACT | HAMMING | CONSISTENT |
| MLP+FIL | 5.6 | 85.9 | 7.0 | 1.0 | **75.8** | 2.7 |
| MLP+$\mathcal{L}_{SL}$ | 28.5 | 83.1 | 75.2 | 15.0 | 72.4 | 69.8 |
| MLP+NESYENT | 30.1 | 83.0 | 91.6 | 18.2 | 71.5 | 96.0 |
| MLP+SPL (*ours*) | **37.6** | **88.5** | **100.0** | **20.8** | 72.4 | **100.0** |

**Theorem 3.1** (Efficient inference in SPLs). *If $q(\mathbf{Y}; \boldsymbol{\Theta})$ and $c_K(\mathbf{Y}, \mathbf{X})$ are two smooth, decomposable and compatible circuits, then computing Eq. (2) can be done in $\mathcal{O}(|q||c|)$ time. Furthermore, if they are also deterministic, then computing the MAP state can be done in $\mathcal{O}(|q||c|)$ time.*

The proof can be found in Sec. A. How do we come up with compatible circuits? One option is to have a PC $q$ that is compatible with every possible smooth and decomposable circuit $c$. To do so, we can represent $q$ as a mixtures of $M$ fully-independent models; i.e., $\sum_{i=1}^{M} \omega_i \prod_j q(Y_j; \boldsymbol{\Theta}_i)$. This additional sum unit can be enough to be more expressive than a FIL and already delivers more accurate predictions than any competitor, as our experiments in pathfinding show (Sec. 5). An example of such a circuit is shown in Fig. 3. Another sufficient condition for compatibility is that both $q$ and $c$ share the exact same hierarchical scope partitioning [75], sometimes called a vtree or variable ordering [13, 59].

This can be done easily if one first compiles logical constraints into OBDDs or SDDs and then uses a mechanized algorithm to build $q$ as in [58] to create a compatible structure. Additionally, to ensure $q$ is a deterministic PC, we could exploit the mechanized construction proposed in Shih and Ermon [69]. Computing the exact MAP state, however, is of less concern as approximate inference algorithms, e.g., beam search decoding [79] or iterative pruning [12], are nowadays a commodity in deep learning frameworks. For non-deterministic PCs, we compute the MAP state with a faster approximation by replacing non-deterministic sum units with max units [55]. This runs in time linear in the size of $r$, and yet delivers state-of-the-art accuracies in our experiments Sec. 5.

## 3.4 A single-circuit SPL

The two-circuit design we proposed for SPLs provides a clear and theoretically-backed interface between neural networks and probabilistic and symbolic reasoning. This setup, however, can sometimes be wasteful, as it requires to compute the product of two circuits and renormalize. We circumvent this issue by designing a single-circuit implementation of SPL.

**Definition 3.8** (Single-circuit SPL). Given an input configuration $\boldsymbol{x}$, a single-circuit SPL computes $p(\boldsymbol{y} \mid \boldsymbol{x}) = c_K(\mathbf{Y}, \mathbf{X}; \boldsymbol{\Omega} = g(f(\mathbf{X})))$ where $c_K$ is a neural conditional constraint circuit whose sum-unit parameters $\boldsymbol{\Omega}$ are non-unitary values parameterized via a gating function $g$.

In a nutshell, we can directly realize SPL by compiling a complex logical constraints (D4) into a deterministic constraint circuit $c_K$, as before, and then parameterizing it with a gating function of the network embeddings $f(\mathbf{X})$, i.e., allowing its sum units to be non-unitary and input dependent. Since the support of $c_K$ is already restricted to exactly match the constraint K (D3), parameterizing $\boldsymbol{\Omega}$ induces an expressive probability distribution over the label configurations that are consistent with K (D2). We can further guarantee that the circuit's output are normalized probabilities (D1, D6) by enforcing the parameters $\omega$ of each sum unit to form a convex combination [57]. This can be easily done by utilizing a softmax activation function for $g$.

One of the advantages of the two-circuit implementation of SPLs is that the size of the circuit $q_{\boldsymbol{\Theta}}$ can be easily increased to improve the capacity of the model (Sec. 3.1). The single-circuit implementation is not as flexible, as normally the number of parameters is determined by the complexity of the constraint circuit, which depends entirely on the compilation step. In this case, one option is to *overparameterize* the neural conditional circuit by introducing additional sum units, hence allowing it to capture more modes in the distribution. We detail this process is Sec. C. A side effect of overparameterization is that it relaxes determinism, meaning that MAP inference needs to be approximated, as described in Sec. 3.3. Additionally, training a gating function to map relatively small embeddings to large parameter vectors in overparameterized circuits, can slow down training. In such cases, a two-circuit implementation of SPL is to be preferred.

# 4 Related works

In this section, we position SPLs against state-of-the-art approaches for enforcing constraints on neural network predictions. In-depth surveys on this topic can be found in [19] and [34].

**Energy-based models.** Deep energy-based models (EBMs) replace FILs with an unnormalized factor graph [42] that captures higher-order label dependencies [43] (D2) but at the cost of foregoing probabilistic semantics (D1) and efficiency (D6). EBMs are typically unconcerned with hard constraints (D3). Neural approaches for segmentation [47] and parsing [28, 82, 83] remedy to this by replacing the factor graph with a full-fledged intractable (discriminative) graphical model [42]. To gain efficiency, one can restrict EBMs to simpler graphical models (e.g., chains, trees), compromising expressiveness (D2) and the ability to model non-trivial logical constraints (D3, D4).

**Loss-based methods.** A prominent strategy consists of penalizing the network for producing inconsistent predictions using an auxiliary loss [19, 34]. While popular, loss-based methods, however *cannot* guarantee that the predictions will be consistent at test time. Common losses include translating logical constraints into a differentiable fuzzy logic [25, 26], as exemplified by DL2 [29]. Although efficient (D6), this solution is not probabilistically sound (D1) and crucially *is not syntax-invariant*: different encodings of the same formula (e.g., conjunctive vs. disjunctive normal form) yield different losses [31, 24]. Closer to our SPL, the Semantic Loss (SL) [80] avoids this issue by penalizing the the probability $\theta_i$ associated to the $i$-th label by the neural network via the loss term

$$\mathcal{L}_{\mathsf{SL}} \propto - \sum_{y \models \mathsf{K}} \prod_{\boldsymbol{y} \models Y_i} \theta_i \prod_{\boldsymbol{y} \not\models Y_i} (1 - \theta_i) = - \sum_{\boldsymbol{y} \models \mathsf{K}} \prod_i p(Y_i \mid \boldsymbol{x}) = - \sum_{\boldsymbol{y}} \prod_i q(Y_i; \theta_i) \cdot c_{\mathsf{K}}(\boldsymbol{x}, \boldsymbol{y}).$$

When K is compiled into a constraint circuit $c_{\mathsf{K}}$ one retrieves $-\mathcal{Z}(\boldsymbol{x})$ for a two-circuit version of SPL that is as expressive as FIL as it assumes independent labels via a conditional PC $\prod_i q(Y_i; \theta_i)$. The neuro-symbolic entropy (NESYENT) [3] extends $\mathcal{L}_{\mathsf{SL}}$ by an entropy term that improves (but still does not guarantee) consistency. It still makes the same independence assumptions over labels (D2).

**Consistency layers.** Approaches ensuring consistency by embedding the constraints into the predictive layer as in SPLs include MultiplexNet [38] and HMCCN [32]. MultiplexNet is able to encode only constraints in disjunctive normal form, which is problematic for generality (D4) and efficiency (D6) as neuro-symbolic SOP tasks involve an intractably large number of clauses – e.g. our pathfinding experiments involves billions of clauses. HMCCN encodes label dependencies as fuzzy relaxation and is the current state-of-the-art model for HMLC [32]. HMCCN and even its recent extension [33] are restricted to only certain constraints that can be exactly encoded with fuzzy logic easily. SPLs instead can express constraints encoded as arbitrary propositional logical formulas (D4).

**Other approaches.** Other common approaches to neuro-symbolic SOP require to invoke a solver to either obtain the MAP state or to compute (often only approximately) the gradient of the loss [23, 60, 52]. SPLs have no such requirement. Some neuro-symbolic approaches [66] constrain the outputs of neural networks within complex logical reasoning pipelines to solve tasks harder than neuro-symbolic SOP. For instance, DeepProblog [48] uses Prolog's backward chaining algorithm for first order logical rules whose probabilistic weights are predicted by the network. In modern implementations of Problog, grounding a first order program and then compiling it into constraint circuits [27] produces a conditional circuit akin to those we use in SPLs, but in which (*i*) only input distributions are parameterized and (*ii*) increasing the parameter count is not considered a straightforward operation. Scallop [39] provides a more scalable approach to deepproblog by considering only the top-$k$ proofs. We leave to future work how we could quickly compile only a specific query as DeepProblog/Scallop do, to deal with first-order representations efficiently.

# 5 Experiments

We evaluate SPLs on standard neuro-symbolic SOP benchmarks such as *simple path prediction*, *preference learning* [80], *shortest path finding in Warcraft* [60] and *HMLC* [32]. We compare SPLs against several state-of-the-art loss- and layer-based approaches (Sec. 4) by applying them to the same base neural network architecture as feature extractor $f$. As we are interested in measuring how close to the ground truth and how safe the predictions of all models are, we report the percentage of EXACT matches of the predicted labels, also called subset accuracy [74], and the percentage of CONSISTENT predictions, also called "Constraint" [80]. Note that, like other consistency layers,

Table 3: SPLs outperform competitors in pathfinding in Warcraft. Predicted paths that do not exactly match the ground truth are still valid paths and yield very close costs to the ground truth. Competitors' predictions can have higher Hamming scores but be invalid. More examples in Sec. D.3.

| ARCHITECTURE | EXACT | HAMMING | CONSISTENT |
|---|---|---|---|
| RESNET-18+FIL | 55.0 | **97.7** | 56.9 |
| RESNET-18+$\mathcal{L}_{SL}$ | 59.4 | **97.7** | 61.2 |
| RESNET-18+SPL (*ours*) | **78.2** | 96.3 | **100.0** |

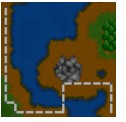 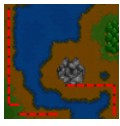 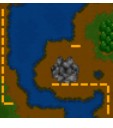 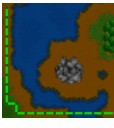

GROUND TRUTH     FIL     $\mathcal{L}_{SL}$     SPL

SPLs are guaranteed to always output $100\%$ consistent predictions. Additionally, we report the HAMMING score [74], mainly to maintain compatibility with previous experimental settings [80, 3]. This metric does not consider consistency of predictions and naturally favors competitors that assume label independence and thus can minimize the per-label cross-entropy [22] (Table 3). Sec. D collects all experimental details such as architectures and hyperparameters used for each experiment.

In Sec. F we provide the average timings for compiling logical formulas into circuits—carried out once, and reused in all subsequent experiments, for parameterizing the conditional circuits, computing the MAP-state of SPL and the loss function at training time (including the cost of computing the product circuit $r$ and its normalization). All these timings, compilation excluded, are reported per batch. We compare to the timings of baselines such as semantic loss and neuro-symbolic entropy, where applicable, to which SPL is highly competitive.

**Simple path prediction & preference learning.** We start by comparing SPLs against loss-based approaches, reproducing the neuro-symbolic benchmarks of Xu et al. [80] for simple path prediction and preference learning. In the first experiment, given a source and destination node in an unweighted grid $G = (V, E)$, the neural net needs to find the shortest unweighted path connecting them. We consider a $4 \times 4$ grid. The input $(\boldsymbol{x}, \boldsymbol{y})$ is a binary vector of length $|V| + |E|$, with the first $|V|$ variables indicating the source and destination nodes, and the subsequent $|E|$ variables indicating a subgraph $G' \subseteq G$. Each label is a binary vector of length $|E|$ encoding the unique shortest path in $G'$. For each example, we obtain $G'$ by dropping one third of the edges in the graph $G$ uniformly at random, filter out the connected components with fewer than 5 nodes, to reduce degenerate cases, and then sample a source and destination node uniformly at random from $G'$. The dataset consists of $1600$ such examples, with a $60/20/20$ train/validation/test split.

In the preference learning task, given a user's ranking over a subset of items, the network has to predict the user's ranking over the remaining items. We encode an ordering over $n$ items as a binary matrix $Y_{ij}$, where for each $i, j \in 1, \ldots, n$, $Y_{ij}$ indicates whether item $i$ is the $j$th element in the ordering. The input $\boldsymbol{x}$ consist of the user's preference over 6 sushi types, and the model has to predict the user's preferences (a strict total order) over the remaining 4. We use preference ranking data over 10 types of sushi for $5,000$ individuals, taken from [49], and a $60/20/20$ split.

We employ a 5-layer and 3-layer MLP as a baseline for the simple path prediction, and preference learning, respectively, equipped with FIL layer and additionally with the Semantic Loss [80] (MLP+$\mathcal{L}_{SL}$) or its entropic extension [3] (MLP+NESYENT). We compile the logical constraints into an SDD [17] and then turn it into a the same constraint circuit $c_K$ that is used for $\mathcal{L}_{SL}$, NESYENT (Sec. 4) and our 1-circuit implementation of SPLs. Table 2 clearly shows that the increased expressiveness of SPL, coming from overparameterizing $c_K$, allows to outperform all competitors while guaranteeing consistent predictions, as expected.

**Warcraft Shortest Path.** Next, we evaluate SPL on the more challenging task of predicting the minimum cost path in a weighted $12 \times 12$ grid imposed over terrain maps of Warcraft II [60]. Each vertex is assigned a cost corresponding to the type of the underlying terrain (e.g., earth has lower cost than water). The minimum cost path between the top left and the bottom right vertices of the grid is encoded as an indicator matrix, and serves as a label. As in [60] we use a ResNet18 [36] with FIL optionally with $\mathcal{L}_{SL}$ as a baseline. Given the largest size of the compiled constraint circuit $c_K$ in this case $10^{10}$, we use a two-circuit implementation of SPL. Results in Fig. 1 and Table 3 are striking: not only SPL outperforms competitors by a large margin – approx. $+23\%$ over FIL and $+19\%$ over the SL – but also consistently delivers meaningful paths that are very close to the ground truth in terms of cost, even when they encode very different routes. See Sec. D.3 for a gallery of these examples. Concerning times, SPLs are able to compute the likelihood in a mere 14 seconds per batch even on a $10^{10}$ valid configuration space (Sec. F).

Table 4: Comparison between SPL and HMCNN [32] on twelve HMLC datasets averaged over 10 runs. Best results for each dataset are in bold. Results which are not significantly worse than the competition, as determined using an unpaired Wilcoxon test, are marked in boldface. Consistency is always 100% for both approaches.

| DATASET | EXACT MATCH | | HAMMING SCORE | |
|---|---|---|---|---|
| | HMCNN | MLP+SPL | HMCNN | MLP+SPL |
| CELLCYCLE | $3.05 \pm 0.11$ | $\mathbf{3.79 \pm 0.18}$ | $\mathbf{98.26 \pm 0.00}$ | $97.84 \pm 0.06$ |
| DERISI | $1.39 \pm 0.47$ | $\mathbf{2.28 \pm 0.23}$ | $\mathbf{98.32 \pm 0.32}$ | $97.70 \pm 0.07$ |
| EISEN | $5.40 \pm 0.15$ | $\mathbf{6.18 \pm 0.33}$ | $\mathbf{98.09 \pm 0.01}$ | $97.30 \pm 0.04$ |
| EXPR | $4.20 \pm 0.21$ | $\mathbf{5.54 \pm 0.36}$ | $\mathbf{98.29 \pm 0.01}$ | $97.87 \pm 0.02$ |
| GASCH1 | $3.48 \pm 0.96$ | $\mathbf{4.65 \pm 0.30}$ | $\mathbf{98.37 \pm 0.31}$ | $97.59 \pm 0.05$ |
| GASCH2 | $3.11 \pm 0.08$ | $\mathbf{3.95 \pm 0.28}$ | $\mathbf{98.27 \pm 0.00}$ | $97.94 \pm 0.07$ |
| SEQ | $5.24 \pm 0.27$ | $\mathbf{7.98 \pm 0.28}$ | $\mathbf{98.31 \pm 0.01}$ | $97.66 \pm 0.03$ |
| SPO | $\mathbf{1.97 \pm 0.06}$ | $\mathbf{1.92 \pm 0.11}$ | $\mathbf{98.23 \pm 0.00}$ | $98.17 \pm 0.03$ |
| DIATOMS | $48.21 \pm 0.57$ | $\mathbf{58.71 \pm 0.68}$ | $\mathbf{99.75 \pm 0.00}$ | $99.64 \pm 0.01$ |
| ENRON | $5.97 \pm 0.56$ | $\mathbf{8.18 \pm 0.68}$ | $\mathbf{94.10 \pm 0.04}$ | $93.19 \pm 0.13$ |
| IMCLEF07A | $79.75 \pm 0.38$ | $\mathbf{86.08 \pm 0.45}$ | $\mathbf{99.40 \pm 0.01}$ | $99.35 \pm 0.03$ |
| IMCLEF07D | $76.47 \pm 0.35$ | $\mathbf{81.06 \pm 0.68}$ | $98.06 \pm 0.02$ | $\mathbf{98.07 \pm 0.08}$ |

**Hierarchical Multi-Label Classification.** Lastly, we follow the experimental setup of Giunchiglia and Lukasiewicz [32] and evaluate SPL on 12 real-world HMLC tasks spanning four different domains: 8 functional genomics, 2 medical images, 1 microalgea classification, and 1 text categorization. Fig. 3 shows an example of a hierarchy of classes. These tasks are especially challenging due to the limited number of training samples, the large number of output classes, ranging from 56 to 499, as well as the sparsity of the output space. The larger datasets yield a label space of $2^{499}$ configurations, but we can compile them in seconds into compact constraints circuits of size $\approx 108$KB (Sec. F).

For numeric features we replaced missing values by their mean, and for categorical features by a vector of zeros, and standardized all features. We used the validation splits to determine the number of layers in the gating function as well as the overparameterization, keeping all other hyperparameters fixed. The final models were obtained by training using a batch size of 128 and early stopping on the validation set. We compare our single-circuit SPL against HMCNN which was shown to outperform several other state-of-the-art HMLC approaches in Giunchiglia and Lukasiewicz [32]. We study the effect of increasing the expressivenss of SPL via overparameterization in Sec. D.4. The results in Table 5 highlight that SPL significantly outperforms HMCNN in terms of exact match on 11 data sets performing comparably on 1,

# 6   Conclusion

SPLs offer the first clear interface for integrating complex probabilitistic reasoning and logical constraints on top of any neural network classifier while retaining efficient inference and training. They improve by a noticeable margin the current state-of-the-art on challenging neuro-symbolic SOP benchmarks such as pathfinding and HMLC. This opens up a number of interesting research directions. First, SPLs can be extended to incorporate logical constraints over multiple networks and representable by first-order formulas [48], which we plan to explore in future works, making the circuit construction pipeline totally transparent to users [2] while possibly automatically learning constraints from data [21, 50]. Second, we are interested in leveraging SPLs to inject scalable logical constraints into large language models [7] thus equipping them with probabilistic reasoning [30, 81].

## Acknowledgments and Disclosure of Funding

The authors would like to thank Arthur Choi for helpful discussions on compiling the constraints for the Warcraft Shortest Path task, and Andreas Grivas for proofreading a draft manuscript. The research of ST was partially supported by TAILOR, a project funded by EU Horizon 2020 research and innovation programme under GA No 952215. This work was funded in part by the DARPA Perceptually-enabled Task Guidance (PTG) Program under contract number HR00112220005,, NSF grants #IIS-1943641, #IIS-1956441, #CCF-1837129, Samsung, CISCO, and a Sloan Fellowship.

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
