## A  Proofs

**Theorem 3.1** (Efficient inference in SPLs). *If $q(\mathbf{Y}; \boldsymbol{\Theta})$ and $c_{\mathsf{K}}(\mathbf{Y}, \mathbf{X})$ are two smooth, decomposable and compatible circuits, then computing Eq. (2) can be done in $\mathcal{O}(|q||c|)$ time, where $|\cdot|$ denotes the circuit size. Furthermore, if they are also deterministic, then computing the MAP state can be done in $\mathcal{O}(|q||c|)$ time. .*

We prove the first statement by first showing that the partition function $\mathcal{Z}(\boldsymbol{x})$ in Eq. (2) can solved exactly in time $\mathcal{O}(|q||c|)$. It will then follow from it that computing Eq. (2) can be done in $\mathcal{O}(|q||c| + |q| + |c|) \approx \mathcal{O}(|q||c|)$ where the last two additive factors derive from evaluating $q$ and $c$ for an input configuration $(\boldsymbol{x}, \boldsymbol{y})$.

To do so, we will exploit two ingredients: i) the product of $q$ and $c$ can be represented as a smooth and decomposable circuit in time $\mathcal{O}(|q||c|)$ [75] and ii) any smooth and decomposable circuit guarantees tractable marginalization in time linear in its size [13]. The next two propositions formalize these statements.

**Proposition A.1** (Tractable product of circuits). *Let $q(\mathbf{Y}; \boldsymbol{\Theta})$ and $c_{\mathsf{K}}(\mathbf{Y}, \mathbf{X})$ be two smooth, decomposable circuits that are compatible over $\mathbf{Y}$ then computing their product as a circuit $r_{\boldsymbol{\Theta}, \mathsf{K}}(\mathbf{X}, \mathbf{Y}) = q(\mathbf{Y}; \boldsymbol{\Theta}) \cdot c_{\mathsf{K}}(\mathbf{Y}, \mathbf{X})$ that is decomposable over $\mathbf{Y}$ can be done in $\mathcal{O}(|q||c|)$. If both $q$ and $c$ are also deterministic, then $r$ is as well.*

*Proof.* The proof directly follows from Theorem 3.2 from Vergari et al. [75]. $\qquad\square$

Note that $\mathcal{O}(|q||c|)$ is a loose upperbound and the size of $r$ is in practice smaller [75].

**Proposition A.2** (Tractable marginalization of circuits). *Let $r(\mathbf{X}, \mathbf{Y})$ be a circuit that is smooth and decomposable over $\mathbf{Y}$ with input functions over $\mathbf{Y}$ that can be tractably marginalized out. Then for any variables $\mathbf{Y}' \subseteq \mathbf{Y}$ and their assignment $\boldsymbol{y}'$, the marginalization $\sum_{\boldsymbol{y}'} r(\boldsymbol{y}', \boldsymbol{y}'', \boldsymbol{x})$ can be computed exactly in time linear in the size of $r$, where $\mathbf{Y}'' = \mathbf{Y} \setminus \mathbf{Y}'$.*

*Proof.* The proof follows by considering that i) the input functionals in SPLs are simple distributions such as Bernoullis and indicators and can be easily marginalized in $\mathcal{O}(1)$ and ii) that for every configuration $\boldsymbol{x}$ of variables $\mathbf{X}$, $r(\mathbf{Y}, \boldsymbol{x})$ is a circuit only over $\mathbf{Y}$ and therefore Proposition 2.1 from Vergari et al. [75] can be directly applied. $\qquad\square$

Analogously, the second statement of Theorem 3.1 follows from Proposition A.1 and by recalling that the MAP state of a deterministic circuit can be computed in time linear in its size.

**Proposition A.3** (Tractable MAP state of circuits (Choi et al. [13])). *Let $r(\mathbf{X}, \mathbf{Y})$ be a circuit that is smooth and decomposable and deterministic over $\mathbf{Y}$ then for a configuration $\boldsymbol{x}$ its MAP state $\operatorname{argmax}_{\boldsymbol{y}} r(\boldsymbol{x}, \boldsymbol{y})$ can be computed in time $\mathcal{O}(|r|)$.*

## B  Compiling logical formulas into circuits

For our experiments we use standard compilation tools to obtain a constraint circuit starting from a propositional logical formula in conjunctive normal form. Specifically, we use Graphillion[1] to compile the constraints in the Warcraft pathfinding experiment into an SDD. For all other experiments, we use PySDD[2] [1] a python SDD compiler [17, 10].

We now illustrate step-by-step one example of such a compilation for a simple logical formula. Consider the constraint circuit $c$ in Fig. 3 encoding the constraint

$$(Y_{\mathsf{cat}} \implies Y_{\mathsf{animal}}) \wedge (Y_{\mathsf{dog}} \implies Y_{\mathsf{animal}}). \tag{3}$$

Intuitively, our aim is to compile the above logical formula into a *compact* form representing all possible assignments to $Y_{\mathsf{cat}}, Y_{\mathsf{dog}}, Y_{\mathsf{animal}}$ satisfying the above constraint. We compile such a constraint by proceeding in a bottom up fashion, where bottom-up compilation can be seen as composing Boolean sub-functions whose domain is determined by a variable ordering, also called

---

[1]https://github.com/takemaru/graphillion
[2]https://github.com/wannesm/PySDD

vtree (see Sec. 3.3). In this example, we assume the function $f(Y_{\text{animal}}, Y_{\text{cat}}, Y_{\text{dog}})$ decomposes as $f_1(Y_{\text{animal}}) \cdot f_2(Y_{\text{dog}}) \cdot f_3(Y_{\text{cat}})$ We therefore start by compiling a constraint circuit that is a function of $Y_{\text{cat}}$ and $Y_{\text{dog}}$, and compose it with a constraint circuit that is a function of $Y_{\text{animal}}$ We first introduce input functionals representing indicators associated with $Y_{\text{cat}}, Y_{\text{dog}}, Y_{\text{animal}}$. We will denote by $Y_i$ the indicator $\mathbb{1}\{Y_i = 1\}$ and by $\neg Y_i$ the indicator $\mathbb{1}\{Y_i = 0\}$.

$$\mathbb{1}\{Y_1 = 0\} \bigcirc \quad \mathbb{1}\{Y_1 = 1\} \bigcirc \quad \mathbb{1}\{Y_2 = 0\} \bigcirc \quad \mathbb{1}\{Y_2 = 1\} \bigcirc \quad \mathbb{1}\{Y_3 = 0\} \bigcirc \quad \mathbb{1}\{Y_3 = 1\} \bigcirc$$

We start by disjoining the indicator $Y_{\text{cat}}$ with $\neg Y_{\text{cat}}$. This corresponds to introducing deterministic and smooth sum units in our circuits.

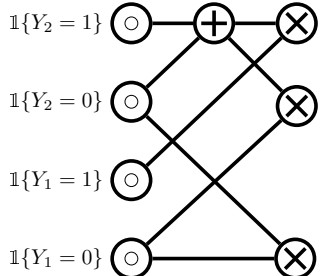

Deterministic sum units represent *disjoint solutions* to the logical formula, meaning there exists distinct assignments, characterized by the children, that satisfy the logical constraint e.g. $Y_{\text{cat}}, Y_{\text{dog}}, Y_{\text{animal}}$ and $\neg Y_{\text{cat}}, Y_{\text{dog}}, Y_{\text{animal}}$ are two distinct assignments which satisfy the constraint.

The compilation process proceeds by conjoining the constraint circuits for $Y_{\text{cat}} \vee \neg Y_{\text{cat}}$ with $Y_{\text{dog}}$, $Y_{\text{cat}} \vee \neg Y_{\text{cat}}$ with $\neg Y_{\text{dog}}$, and $\neg Y_{\text{cat}}$ with $\neg Y_{\text{dog}}$.

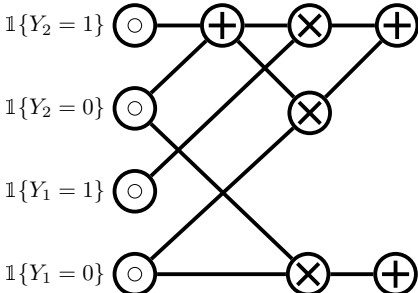

A decomposable product unit *decomposes* functions over disjoint sets of variables. The above products represent the Boolean functions $(Y_{\text{cat}} \vee \neg Y_{\text{cat}}) \wedge Y_{\text{dog}}$, $(Y_{\text{cat}} \vee \neg Y_{\text{cat}}) \wedge \neg Y_{\text{dog}}$, and $\neg Y_{\text{dog}} \wedge \neg Y_{\text{cat}}$.

We disjoin $(Y_{\text{cat}} \vee \neg Y_{\text{cat}}) \wedge Y_{\text{dog}}$ with $(Y_{\text{cat}} \vee \neg Y_{\text{cat}}) \wedge \neg Y_{\text{dog}}$, and $\neg Y_{\text{dog}} \wedge \neg Y_{\text{cat}}$ with true, the logical multiplicative identity, guaranteeing alternating sum and product nodes, as mentioned in Sec. 3.1.

So far, we have compiled constraint circuits for the logical formulas

$$((Y_{\text{cat}} \vee \neg Y_{\text{cat}}) \wedge Y_{\text{dog}}) \vee ((Y_{\text{cat}} \vee \neg Y_{\text{cat}}) \wedge \neg Y_{\text{dog}})) \tag{4}$$

and

$$\neg Y_{\text{dog}} \wedge \neg Y_{\text{cat}}. \tag{5}$$

What remains is to conjoin Eq. (4) with $Y_{\text{animal}}$, and Eq. (5) with $\neg Y_{\text{animal}}$, and disjoin the resulting constraint circuits. What we get is a mixture over the possible solutions: If we predict there is a dog or a cat, or both, in e.g., an image, we better predict that there's an animal. Conversely, the absence of a dog and a cat from an image implies nothing as to the presence of an animal in the image.

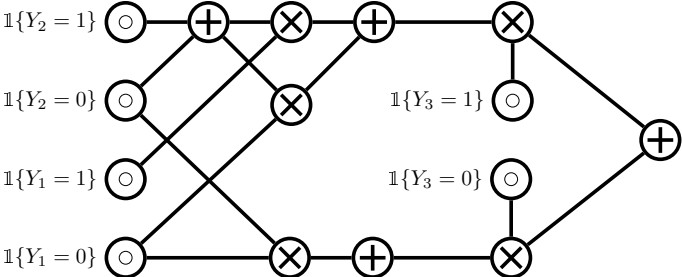

Compilation techniques like the one we illustrated do not, however, escape the hardness of the problem: the compiled circuit can be exponential in the size of the constraint, *in the worst case*. *In practice*, nevertheless, we can obtain compact circuits because real-life logical constraints exhibit enough structure (e.g., they encode repeated sub-problems) that can be easily exploited by a compiler. We refer to the literature of compilation for details on this [18].

## C   Overparameterizing the single-circuit SPL

As mentioned in Def. 3.8, SPLs can be realized as a single circuit by first compiling a complex logical constraint into a deterministic constraint circuit, and then parameterizing it using a gating function of the network embeddings. Intuitively, this parameterization induces a probability distribution over the possible solutions of a logical formula encoded in the constraint circuit. The expressiveness of this distribution depends on the number of parameters of the constraint circuit, i.e., the number of weighted edges associated to sum units. As we would like to endow our single-circuit SPL with the ability to induce complex distributions, we devise *two strategies* to introduce more parameters than what the constraint circuit alone can offer: *replication* and *mixture multiplication*.

Replication works by maintaining $m$ copies of the circuit, and taking their weighted average, i.e., introducing a sum unit that mixes them [58]. Mixture multiplication, instead, substitutes a single local marginal distribution encoded by a sub-circuit rooted into a sum unit with $k$ mixture models over the same scope. In practice, we create $k-1$ copies of each sum units and rewire them by computing a cross product of their inputs as in Peharz et al. [58]. Algorithm 1 formalizes this process.

As mentioned in Def. 3.8, both strategies relax determinism. However, note that *they do not alter the support of the underlying distribution*. This guarantees that all the predictions will be consistent with the encoded constraint (D3) (Sec. 2).

## D   Additional experimental details

### D.1   Simple path prediction and preference learning

In the simple path prediction task, given a source and destination node in an unweighted grid $G = (V, E)$, the neural net needs to find the shortest unweighted path connecting them. We consider a $4 \times 4$ grid. The input $(\boldsymbol{x}, \boldsymbol{y})$ is a binary vector of length $|V| + |E|$, with the first $|V|$ variables indicating the source and destination nodes, and the subsequent $|E|$ variables indicating a subgraph $G' \subseteq G$. Each label is a binary vector of length $|E|$ encoding the unique shortest path in $G'$. For each example, we obtain $G'$ by dropping one third of the edges in the graph $G$ uniformly at random, filtering out the connected components with fewer than 5 nodes, to reduce degenerate cases, and then sample a source and destination node uniformly at random from $G'$. The dataset consists of 1600 such examples, with a $60/20/20$ train/validation/test split.

In the preference learning task, given a user's ranking over a subset of items, the network has to predict the user's ranking over the remaining items. We encode an ordering over $n$ items as a binary matrix $Y_{ij}$, where for each $i, j \in 1, \ldots, n$, $Y_{ij}$ indicates whether item $i$ is the $j$th element in the ordering. The input $\boldsymbol{x}$ consist of the user's preference over 6 sushi types, and the model has to predict the user's preferences (a strict total order) over the remaining 4. We use preference ranking data over 10 types of sushi for $5,000$ individuals, taken from [49], and a 60/20/20 split.

We follow Xu et al. [80] in employing a 5-layer with 50 hidden units each and sigmoid activation functions, and 3-layer MLP with 50 hidden units each as a baseline for the simple path prediction,

---

**Algorithm 1** OVERPARAMETERIZE($c, k$, cache, first_call)

---

1: **Input:** a smooth, deterministic, and structured-decomposable circuit $c$ over variables $\mathbf{X}$, an overparameterization factor $k$, and a cache for memoization, and a flag to denote the first call
2: **Output:** an overparameterized, smooth, and structured-decomposable circuit $c$ over $\mathbf{X}$
3: **if** $q \in$ cache **then**
4:     **return** cache $[q]$
5: **if** $c$ is an input unit **then**
6:     nodes $\leftarrow [c]$
7: **else if** $c$ is a sum unit **then**
8:     elements $\leftarrow [\,]$
9:     //For every product unit that is an input of $c$
10:    //recursively overparameterize its inputs,
11:    //which are sum units, and take their cross (cartesian) product
12:    **for** $(c_L, c_R) \in$ in$(c)$ **do**
13:        left $\leftarrow$ OVERPARAMETERIZE$(c_L, k)$
14:        right $\leftarrow$ OVERPARAMETERIZE$(c_R, k)$
15:        elements.APPEND$([$CROSSPRODUCT$($left, right$)]$)
16:    in$(c) \leftarrow$ elements
17:    nodes $= [c] + [$COPY$(c)$ **for** $i = 1$ **to** $k]$
18: **if** first_call **then**
19:    //Create a sum unit whose inputs are nodes
20:    //and whose parameters are 1s.
21:    nodes $\leftarrow$ SUM$($nodes, $\{1\}_{i=1}^{|\mathsf{nodes}|})$
22: cache$(c) \leftarrow$ nodes
23: **return** nodes

---

and preference learning, respectively. We equip this baselines with a FIL and additionally with the Semantic Loss [80] (MLP+$\mathcal{L}_{\mathsf{SL}}$) or its entropic extension [3] (MLP+NESYENT).

We compile the logical constraints into an SDD [17] and then turn it into a constraint circuit $c_{\mathsf{K}}$ that is used for $\mathcal{L}_{\mathsf{SL}}$, NESYENT (Sec. 4) and our 1-circuit implementation of SPLs. To obtain the results for SPL in Table 2, we perform a grid search over the using the validation set for a maximum of 2000 iterations, similar to Xu et al. [80]. We search over the learning rates in the range $\{1 \times 10^{-3}, 5 \times 10^{-3}, 1 \times 10^{-4}, 5 \times 10^{-4}\}$, the overparameterization factor $k$ in the range $\{2, 4, 8\}$, as well as the number of circuit mixtures $m$ in the range $\{2, 4, 8\}$, evaluating the model with the best performance on the validation set.

### D.2 Hierarchical Multi-Label Classification

We follow the experimental setup of Giunchiglia and Lukasiewicz [32] and evaluate SPL on 12 real-world HMLC tasks spanning four different domains: 8 functional genomics, 2 medical images, 1 microalgea classification, and 1 text categorization. These tasks are especially challenging due to the limited number of training samples, the large number of output classes, ranging from 56 to 4130, as well as the sparsity of the output space. We used the same train-validation-test splits and experimental setup as [32]. For numeric features we replaced missing values by their mean, and for categorical features by a vector of zeros, and standardized all features. We used the validation splits to determine the number of layers in the gating function in the range $\{2, 4, 8\}$, the overparameterization factor in the range $\{2, 4, 8\}$, and the number of mixtures in the range $\{2, 4, 8\}$, keeping all other hyperparameters fixed. The final models were obtained by training using a batch size of 128 and early stopping with a patience of 20 on the validation set.

### D.3 Warcraft pathfinding

We evaluate SPL on the more challenging task of predicting the minimum cost path in a weighted $12 \times 12$ grid imposed over terrain maps of Warcraft II [60]. Our setting differs from the one proposed by Pogančić et al. [60] in two ways: i) a node only neighbors four nodes as instead of eight, excluding the diagonals; ii) the neural network predicts the edges in the path, as opposed to the vertices, resolving ambiguities in the previous task (note that a set of vertices can *might* ambiguously encode

Table 5: A comparison of the performance of single-circuit SPL with different parameters: $m$, the number of circuit copies in our replication strategy; *gates*, the number of layers in the gating function; and $k$ the overparameterization factor in the mixture multiplication strategy (Algorithm 1). We report the percentage of exact matches of the predicted labels on the validation set of the *HMLC* dataset, highlighting the best numbers in **boldface**. As can be seen, all datasets benefit from overparameterization.

| DATASET | $m$: 2 | | | | $m$: 4 | | | | $m$: 8 | | | |
| --- | --- | --- | --- | --- | --- | --- | --- | --- | --- | --- | --- | --- |
| | GATES: 2 | | GATES: 4 | | GATES: 2 | | GATES: 4 | | GATES: 2 | | GATES: 4 | |
| | $k$: 2 | $k$: 4 | $k$: 2 | $k$: 4 | $k$: 2 | $k$: 4 | $k$: 2 | $k$: 4 | $k$: 2 | $k$: 4 | $k$: 2 | $k$: 4 |
| CELLCYCLE | 4.25 | 4.48 | 4.48 | 4.01 | 4.60 | **4.83** | 4.25 | 4.48 | 4.36 | 4.13 | 4.36 | 4.13 |
| DERISI | 2.26 | 2.02 | 2.14 | 2.26 | **2.49** | 2.26 | 2.38 | 2.38 | **2.49** | 2.38 | 2.26 | **2.49** |
| EISEN | 6.05 | 6.05 | 6.05 | 6.05 | 5.86 | 6.43 | **6.81** | 6.24 | 6.43 | 6.43 | 6.05 | 6.43 |
| EXPR | 5.42 | 4.83 | 5.18 | 5.30 | 4.83 | **5.54** | **5.54** | 5.18 | **5.54** | 5.42 | 5.18 | 5.42 |
| GASCH1 | 5.56 | 5.79 | 5.67 | 5.91 | 5.44 | 5.67 | 6.03 | **6.26** | 5.79 | 5.79 | **6.26** | 6.03 |
| GASCH2 | 4.00 | 4.24 | 4.83 | **4.95** | 4.12 | 4.00 | 4.12 | 4.36 | 4.24 | 3.53 | 4.24 | 4.59 |
| SEQ | 7.74 | 7.74 | 7.51 | 7.85 | 8.19 | 7.28 | 7.96 | 7.17 | 7.96 | 7.39 | 7.51 | **8.42** |
| SPO | 2.27 | 2.15 | 2.15 | 2.51 | 2.39 | 2.27 | 2.51 | 2.51 | **2.87** | 2.27 | 2.39 | 2.63 |
| DIATOMS | 53.71 | **54.68** | 50.16 | 51.29 | 53.23 | 52.10 | 49.35 | 48.23 | 52.90 | 52.58 | 46.61 | 47.26 |
| ENRON | 19.53 | 18.52 | 17.85 | 19.87 | 19.87 | 20.20 | **20.54** | 20.20 | 19.53 | 20.20 | 19.53 | 19.87 |
| IMCLEF07A | 86.97 | 87.03 | 86.27 | 86.60 | 87.00 | **87.33** | 86.50 | 86.70 | 87.07 | 86.90 | 87.00 | 86.83 |
| IMCLEF07D | 85.93 | 85.80 | 85.87 | 85.73 | 85.60 | **86.50** | 85.87 | 85.90 | 85.87 | 85.83 | 86.10 | 85.50 |

more than one path). Each vertex is assigned a cost corresponding to the type of the underlying terrain (e.g., earth has lower cost than water). The minimum cost path between the top left and the bottom right vertices of the grid is encoded as an indicator matrix, and serves as a label.

We use Graphillion[3] to compile the path constraint, limiting our constraint to the set of paths whose length is less than 29, as determined on the training set.

As in [60] we use a ResNet18 [36] with FIL optionally with $\mathcal{L}_{\mathsf{SL}}$ as a baseline. Given the largest size of the compiled constraint circuit $c_{\mathsf{K}}$ in this case $10^{10}$, we use a two-circuit implementation of SPL. We use the identity function as our gating function and do a grid search over only the number of mixtures in the range $\{2, 4, 8\}$ in our model, keeping all other hyperparameters as proposed in [60].

### D.4 A study on the effect of overparameterization in SPL

We now illustrate the effect that overparameterization has on the performance of the single-circuit SPL. To that end, we performed an ablation study, comparing single-circuit SPLs comprising a different number of circuit copies $m$ for our replication strategy, a different number of layers in the gating function, denoted by *Gates*, and the overparameterization factor $k$ as used in Algorithm 1 in our mixture multiplication strategy.

We report the exact match percentage of the predicted labels on the validation set of the 12 HMLC datasets in Table 5. As a general trend, we can see that our overparameterization strategies pay off and in general more mixture nodes help ($k = 4$) as well as using more replicas ($m \geq 4$). The effect of employing a deeper gating function is less striking instead, with a two-layer gating function achieving highest performances on 9 datasets.

## E  Ethical Considerations

SPLs are meant as a module to be added on top of neural networks, and as such it does not significantly alter the ethical risk of the underlying model and target application. One exception is if the symbolic constraint is wrong (because e.g. it was encoded by a non-expert), in which case enforcing consistency - as SPLs do - may lead to mistakes or bias in the model's predictions.

---

[3]https://github.com/takemaru/graphillion

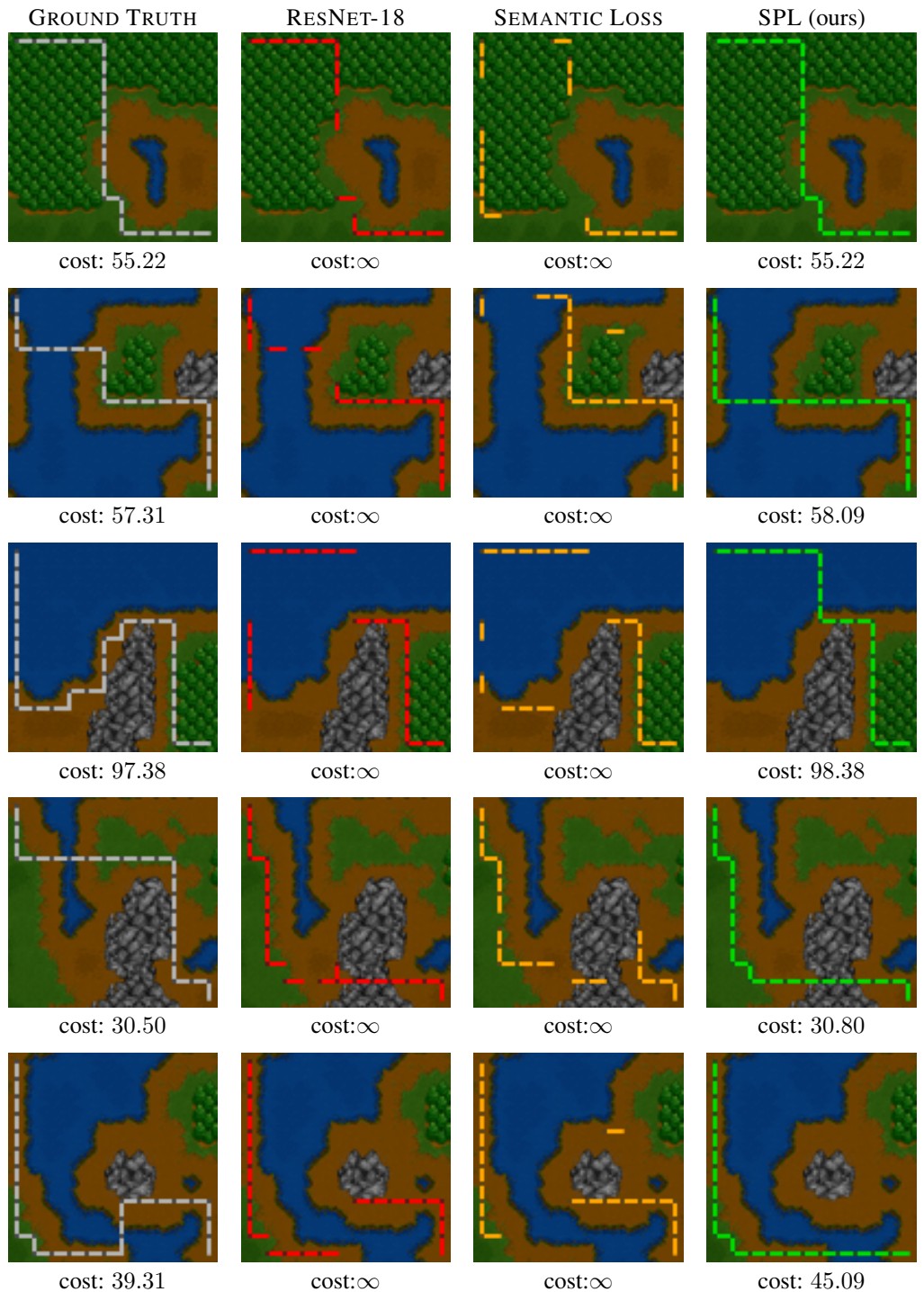

Figure 4: **More examples of shortest path predictions in SPLs and competitors.** SPLs always deliver valid paths and even when these do not exact match the ground truth, they are very close in terms of their global cost. Paths from the baselines might yield a higher Hamming score (as they have more overlapping edges with the ground truth) but are invalid.

# F Timings

Table 6: A comparison of the timings of the different methods used throughout our experiments. All timings are in seconds. The timings for HMLC datasets are obtained by averaging over the timings of an entire epoch. All other timings are the average over three function calls. An empty cell, denoted by a dash, indicates the method was not used for that dataset, and therefore its timing is unavailable.

| DATASET | COMPILATION | $\mathcal{L}_{SL}$ | NESYENT | SPLs | | |
| --- | --- | --- | --- | --- | --- | --- |
| | | | | PARAMETERIZE | CROSS-ENTROPY | MAP |
| CELLCYCLE | 68 | - | - | 0.03 | 0.41 | 0.74 |
| DERISI | 68 | - | - | 0.01 | 0.21 | 0.37 |
| EISEN | 29 | - | - | 0.01 | 0.16 | 0.28 |
| EXPR | 68 | - | - | 0.00 | 0.11 | 0.19 |
| GASCH1 | 68 | - | - | 0.02 | 0.42 | 0.77 |
| GASCH2 | 68 | - | - | 0.03 | 0.40 | 0.74 |
| SEQ | 66 | - | - | 0.01 | 0.22 | 0.36 |
| SPO | 67 | - | - | 0.03 | 0.40 | 0.74 |
| DIATOMS | 8 | - | - | 0.00 | 0.09 | 0.14 |
| ENRON | 0.04 | - | - | 0.01 | 0.16 | 0.28 |
| IMCLEF07A | 0.35 | - | - | 0.00 | 0.06 | 0.11 |
| IMCLEF07D | 0.08 | - | - | 0.00 | 0.05 | 0.10 |
| WARCRAFT | 457 | 16.30 | - | 0.21 | 14.11 | 15.59 |
| PREFERENCE | [80] | 0.024 | 0.035 | 0.00 | 0.00 | 0.01 |
| SIMPLE PATH | [80] | 0.34 | 0.49 | 0.00 | 0.13 | 0.19 |