# OpenReview forum: "Semantic Probabilistic Layers for Neuro-Symbolic Learning"
_NeurIPS.cc/2022/Conference — NeurIPS 2022 Accept_

### Official Review · Reviewer_KRqF · 2022-07-10

**Rating:** 5
**Confidence:** 3
**Soundness:** 3 good
**Presentation:** 2 fair
**Contribution:** 3 good

**Summary:**

This study proposed a differentiable Semantic Probabilistic Layer (SPL) for structure prediction problems. The SPL can be used together with any off-the-shelf neural networks to model conditional distributions p(y|x) where there is additional hard constraints imposed on y. Since the SPL is built upon smooth and decomposable probabilistic circuits, it can be used to compute the probability p(y|x) efficiently in linear time. Such property enables the maximum-likelihood learning of the model parameters. The SPL is empirically demonstrated to outperform its alternatives (mainly semantic loss methods) on several structure prediction benchmarks.

**Questions:**

See *Weaknesses* section for questions.

**Limitations:**

See *Weaknesses* section for limitations.

**Strengths And Weaknesses:**

Strengths:

This paper provided a **generic** and **novel** solution to bridge between neural networks and structure prediction problems. The proposed SPLs can be used to model conditional distributions p(y|x) with logical constraints imposed on y. Backed by smooth and decomposable probabilistic circuits, these SPLs can be used to do exact inference **efficiently**, which enables maximum-likelihood learning over model parameters. The paper provided **theoretical guarantees** for the proposed SPLs and empirically justified them by showing their superior performance over existing methods with semantic loss.

Weaknesses:

The paper is not very self-contained, and it is hard to follow by readers without the background of probabilistic circuits (PCs). In particular the writing of the paper can be improved in the following places:
- "gating function" used without defined in Definition 3.4. Is it similar to the gating functions in RNNs? Why would we want such functions here?
- "... if any pair of product units n\in p and m\in m ..." in Definition 3.7. What is p and m here? How to intuitively understand the compatibility defined here?
- Figure 3 is very difficult to comprehend without furture explanation. What are the inputs and outputs of each unit? What do the red lines refer to? What are the meanings of the conjuctions between the red lines and the black lines? Why are these two circuits compatible?

The computation of the partition function Z(x) in Equation (2) seems to be a time-consuming process. For a different y, we need to compute its c_K(x, y) and use it to weight q_\theta(y | f(x)). This suggest that we need to enumerate over all possible ys in order to compute the normalization term. However, line 122 suggests that we can compute Z(x) in time linea in the size of r_\theta,K. Correct me if I am wrong, if y \in {0,1}^d, isn't that |r_\theta,K| = O(2^d)? Can we still claim that this is efficient?

Regarding the evaluation, the size of the experiment somehow suggests that the proposed approach is not very scalable (e,g, 4x4 path finding experiments). The HMLC experiment seems to be an experiment with decent difficulty, but the baselines are not comprehensive. It could provide a stronger evidence if the authors can compare their method with more methods listed in Table 1 in the experiments.

Suggestions for larger scale evaluation tasks with hard constraints:
1. Sudoku experiment in SATnet[1].
2. Common sense VQA experiment in Scallop[2].
3. RNA Secondary Structure Prediction experiment[3].

In particular, Scallop[2] is a related work which makes deepproblog more scalable.


[1] Wang, Po-Wei, et al. "Satnet: Bridging deep learning and logical reasoning using a differentiable satisfiability solver." International Conference on Machine Learning. PMLR, 2019.

[2] Huang, Jiani, et al. "Scallop: From probabilistic deductive databases to scalable differentiable reasoning." Advances in Neural Information Processing Systems 34 (2021): 25134-25145.

[3] Chen, Xinshi, et al. "RNA Secondary Structure Prediction By Learning Unrolled Algorithms." International Conference on Learning Representations. 2019.

---

> ### Author Response · Authors · 2022-08-02
> **Response to Reviewer KRqF part 1**
>
> We thank the reviewer for their time spent reviewing the paper and for the provided criticism.
> We believe there are some misconceptions about the scalability of SPL and the size of our experiments, which we are keen on clarifying in the comments below. We believe that some minor modifications in the updated version of the pdf can resolve these issues. We hope that these clarifications can lead to an acceptance.
>
> *“The computation of the partition function Z(x) in Equation (2) seems to be a time-consuming process… Can we still claim that this is efficient?”*
>
> The computation of the partition function in SPL is efficient, as practically demonstrated by the computation times that **we have now reported in Appendix F**. See also our answer to reviewer MBwP.
>
> Theoretically, this is possible because, as stated in our Theorem 3.1, the partition function can be computed in time linear in the size of $r$ for a 2-circuit SPL and linear in the size of $c$ for the 1-circuit implementation of SPL. The size of $r$ is $\mathcal{O}(|c|||q)$, which is linear in the sizes of $c$ and $q$. (Note that this is a very loose upper bound as noted in [71], and in practice the size is much smaller). Now, the size of $q$ depends on the learning algorithm used (e.g. [54,73]) or the architecture imposed (e.g., the small mixture model in our Warcraft experiment).
>
> The size of $c$, instead, depends on the knowledge compilation process employed (see Appendix B). In practice, the size of our circuits was compact: 4 KB for the preference learning,  60KB for simple paths, a maximum of 108KB for HMLC and 13MB for Warcraft pathfinding.
>
> *"Small scale experiments"*
>
> We presented larger-scale experiments than the 4x4 grid (which we report for comparing against a large literature on neuro-symbolic losses). In particular, the Warcraft experiment requires learning a distribution over a 12x12 grid, which implies compiling a logical constraint over 10^10 possible different paths. Note that we can do this in less than 14 seconds (see Appendix F and our answer to reviewer MBwP).
>
> *"Comparing over other datasets/tasks:"*
>
> We would like to point out that in this work we are specifically focusing on constraints in propositional logic over binary labels for SOP (as noted in lines 60-72). Some proposed tasks such as the VQA involve whole external knowledge bases and deviate from our semantic SOP setting. It is not clear to us which constraints can be cast only on the scene graph (the output of the neural net) and how not to perform multi-hop reasoning. Even if there is a reduction to our setting, comparing against DeepProblog and Scallop, which operate on a first-order formalism adopt sophisticated ways to compile only a query, and can access external algorithms to reason over a knowledge base (e.g. forward/backward chaining) would be unfair.
>
> In fact,  to cast one logical constraint representing a first-order knowledge base in the propositional language of SPL, we would require to propositionalize it as a whole, with the potential cost of greatly enlarging the state space. We leave to future work an investigation of how we could quickly compile not a whole logic base (as we are doing) but only a specific query (as DeepProblog/Scallop do) in SPL, extending SPL to first-order templates for constraints.
>
> Concerning the HMLC experiments, we point out that the aim of that experiment is comparing against other state-of-the-art consistency layers. Therefore, we are not comparing against semantic loss and its variants (we demonstrate we outperform them in the previous experiments). Also, we cannot compare against MultiplexNet because, as discussed in the related works (lines 283-285), the DNF formalism of Multiplexnet won’t scale with the large number of constraints encoded in the hierarchies of these datasets. For an analogous efficiency reason, we are not comparing against EBM, as they do not support the efficient computation of the exact partition function. Lastly, we do not compare against a simple FIL baseline as it does not guarantee the satisfaction of the constraint. We did not report the baselines used in [31] because they are outperformed by HMCNN, the competitor we used.
>
> *"Additional references"*
>
> We updated the manuscript by referencing the suggested works, pointing out the conceptual differences with SPL as we just discussed.

---

> > ### Comment · Reviewer_KRqF · 2022-08-08
> > **Thank you for the response**
> >
> > Thank you for adding the wall clock time to the Appendix. It is impressive that doing MAP on the warcraft example only takes 14 seconds!
> >
> > Theoretically, I agree that the bottleneck of the size of $r$ is $|c|$, and I understand that after compression, $|c|$ could be relatively small. However, when we think about the worst-case scenario, let's say for $y \in \{0,1\}^d$, isn't it that $|c| = O(2^d)$? Maybe here, instead of claiming that it is efficient all the time, we should say that it is efficient most of the time.
> >
> > I am still considering the experiments somewhat toyish, but it does make sense that the warcraft example has a decent search space.
> >
> > I am increasing my rating since it seems that scalability is not a big issue in practice.

---

> > > ### Author Response · Authors · 2022-08-08
> > > **Response to additional questions and concerns**
> > >
> > > We thank the reviewer for their answers and increasing the score. We briefly follow up with the hope of reaching full acceptance.
> > >
> > > *"Isn't the worst-case scenario $O(2^d)$"*
> > >
> > > The worst-case size of the circuits depends on i) the algorithm employed for compilation and ii) the structure of the constraints, more than the number of labels per se.
> > >
> > > For example, in the Warcraft experiment, we have a label configuration space of $2^{12^2}=2^{144}\approx 10^{43}$ states but only $10^10$ configurations are valid. If our compilation algorithm was simply enumerating these configurations and putting them in a logical OR (as MultiplexNet actually does), $|c|$ would be $10^{10}$. In our experiments, we use the `pySDD` compiler, which can generate circuits whose size can be worst-case exponential in the treewidth of the CNF representation of the constraint.
> > >
> > > We will update the camera-ready with this information.
> > >
> > > *"I am still considering the experiments somewhat toyish"*
> > >
> > > We politely disagree: we adopt 12 real-world datasets, from domains ranging from functional genomics to text categorization. Some of these biological datasets have 499 labels, yielding a target space of 2**499 configurations (and as discussed above with SPL we can compile them in seconds into compact circuits of size 108KB).
> > >
> > > Thanks again for engaging with us, and entertaining our responses.

---

> > > > ### Comment · Reviewer_KRqF · 2022-08-09
> > > > **Thanks for the prompt response**
> > > >
> > > > Thanks for addressing the complexity problem in more detail and incorporating the discussion into the revision. This discussion would be particularly helpful for the readers outside the subfield to understand the method. I really appreciate the efforts! For the experiments, again, I acknowledge that some of the tasks have a large search space, but the paper would benefit from a set of more realistic experiments.

---

> ### Author Response · Authors · 2022-08-02
> **Response to Reviewer KRqF part 2**
>
> *"gating function used without defined"*
>
> We have rephrased the definition to highlight that we name g, the function mapping embedding to circuit parameters, as the gating function. There is no additional constraint in defining nor implementing it, in theory. In practice, we used simple MLPs; Please see the revised definition 3.4 and the paragraph that follows, discussing the choice of gating function.
>
> *"How to intuitively understand the compatibility defined here?"*
>
> An imprecise, but more intuitive definition of compatibility  is: “any two product units with the same scope can decompose in the same way". Consider the first two product units of $q$ and $c$, right to left and top to bottom. Both units follow the same order of decomposition, decomposing $\{Y_3, Y_2, Y_1}$ into ${Y_3}$ and ${Y_2, Y_1}$.
>
> *"What are the inputs and outputs of each unit? What do the red lines refer to? What are the meanings of the conjuctions between the red lines and the black lines? Why are these two circuits compatible?"*
>
> The red dots at the end of the lines represent the outputs of the gating function g, specifically, the set of parameters of the circuit $q$: the parameters $\boldsymbol{\omega}$ attached to the sum units edges and the parameters $\boldsymbol{\lambda}$ attached to the input units (see definition 3.1). **We made this explicit in the new caption.** As described in definition 3.4, in fact, a gating function simply maps the embedding $\mathbf{Z}$ to the set of parameters $\boldsymbol{\Theta}=\{\boldsymbol{\omega},\boldsymbol{\lambda}\}$.

---

> > ### Comment · Reviewer_KRqF · 2022-08-08
> > **Thank you for the response**
> >
> > Thank you for addressing these issues in the revision. Now it makes more sense to me.
> >
> > Additional question: is $m \in m$ in line 212 a typo? What does it mean?

---

### Official Review · Reviewer_MBwP · 2022-07-12

**Rating:** 7
**Confidence:** 2
**Soundness:** 3 good
**Presentation:** 3 good
**Contribution:** 3 good

**Summary:**

This paper presents a method for structured-output prediction that allows for encoding of hard constraints, and learning of soft constraints. Specifically, the distribution p(y|f(x)) is explicitly modelled as a product, where one term of the product is parameterized by a neural network, and the other term is set to 0 when the label configuration fails the hard constraints. To make predictions tractable, circuits are used. Authors experiment in path prediction and hierarchical multi-label classification settings.

**Questions:**

What are the computational considerations in using this approach? A lot of emphasis is put on making sure that tractable learning and inference, so computational comparisons would be helpful.


**Limitations:**

Author mentions that limitations and ethical discussions are in the supplementary materials, but I couldn’t find it there.


**Strengths And Weaknesses:**

The idea is new to the best of my knowledge, and structured output prediction is a topic of interest to the NeurIPS community.

The submission appears technically sound and grounded theoretically---although I am not an expert in this area and could be missing something.

The submission was easy to follow, though I do have a few minor concerns:

- In section 2, the terms “Expressive” and “General” were not defined in a way I expected. I’m still not confident that I understood the author’s intentions, but my guess is that “Expressive” means “allows soft constraints” or learned patterns in the output labels, and that “General” means “allows hard constraints”. I’m not quite sure what the difference is between “General” and “Consistent”?
- A bit nitpicky but the Figure 1 and the Table 3 figure do not print well in grayscale.

---

> ### Author Response · Authors · 2022-08-02
> **Response to Reviewer MBwP**
>
> We would like to thank the reviewer for their thoughtful, and thorough feedback. We will aim to address their concerns below.
>
> *"In section 2, the terms “Expressive” and “General” were not defined in a way I expected. I’m still not confident that I understood the author’s intentions… I’m not quite sure what the difference is between “General” and “Consistent?"*
>
> Intuitively, generality refers to how rich the constraints can be, while expressivity refers to how rich the distribution over the predictions can be.
>
> By Expressive we mean the ability to capture arbitrarily-complex probability distributions over the space of consistent predictions.
> For example, a FIL layer can only capture factorized distributions, while SPL rich multimodal distributions.
>
> On the other hand, General refers to the class of constraints supported by the model considered. SPLs are general in that they support the representation of any propositional logical constraint. On the other hand, the work of Giunchiglia is restricted to hierarchical constraints.
>
> Lastly, Consistent refers to whether the model always guarantees the consistency of the predictions w.r.t. the symbolic constraint. A method such as the semantic loss can be general as they can encode as complex constraints as SPL, but it is not consistent, as it cannot guarantee that all predictions of the neural net at test time are consistent with the constraint.
>
> *"Figure 1 and the Table 3 figure do not print well in grayscale."*
>
> Thank you for pointing it out! We will redraw them to be grayscale- and colorblind-friendly.
>
> *"Computational considerations in using this approach"*
>
> In appendix F we provide the average timings for compilation—carried out once, and reused in all subsequent experiments, parameterization, MAP-state computation and cross-entropy (product + normalization). All these timings, compilation excluded, are per batch. We compare to the timings of semantic loss and neuro-symbolic entropy, where applicable. We point out that the Warcraft experiment—by far the most time consuming—is a distribution over 10^10 possible paths, and yet we’re able to compute the likelihood in a mere 14 seconds per batch.
>
> *"Author mentions that limitations and ethical discussions are in the supplementary materials, but I couldn’t find it there."*
>
> SPLs are meant as a module to be added on top of neural networks, and as such it does not significantly alter the ethical risk of the underlying model and target application.  One exception is if the symbolic constraint is wrong (because e.g. it was encoded by a non-expert), in which case enforcing consistency - as SPLs do - may lead to mistakes or bias in the model's predictions.  We revised the appendix to include ethical considerations. Please see appendix E.

---

> > ### Comment · Reviewer_MBwP · 2022-08-09
> > **Thank you for the response**
> >
> > Thank you for the explanation of the terms in Section 2. I also read the two new appendices and the other reviews.
> >
> > The computational consideration is addressed in the appendix, and I adjusted my rating accordingly.
> >
> > The definition for the terms in Section 2 is much more clear in your explanation, and it makes sense to me now why you chose those terms.

---

> ### Author Response · Authors · 2022-08-08
> **Follow up to rebuttal**
>
> Dear Reviewer,
>
> we would like to follow up to see if our response and the added times address your concerns or if you have further questions. We would really appreciate the opportunity to discuss your assessment further and move towards full acceptance.
>
> Thank you again.
>
> The authors

---

### Official Review · Reviewer_V2bR · 2022-07-12

**Rating:** 8
**Confidence:** 3
**Soundness:** 4 excellent
**Presentation:** 4 excellent
**Contribution:** 3 good

**Summary:**

This paper targets the problem of neural models trained with logical constraints, while performing well in downstream tasks, still lack of proper interface to put hard constraint such that outputs are strictly consistent. To this end, this paper puts a list of 6 ideal necessities for a constrained neural models and proposed semantic probabilistic layer (SPL) that does efficient training and inference. Experiment results span across synthetic tasks and realistic ones, showing strong task performances while guarantee output space structural consistency.

**Questions:**

1. What is the take on the metric of Hamming score. Is this paper suggesting it is a rough metric that does not reflect structural recall of the ground truth and should be avoided in future?

**Ethics Review Area:**

["I don’t know"]

**Limitations:**

I do not see big limitation that could offset the contributions of this work.

**Strengths And Weaknesses:**

1. A theoretical discussion on ideal necessities for constrained neural models.

2. Naturally SPL is a marginal inference layer that is capable of differentiable learning and outputing perfectly consistent structure. The convesion to circuits can be a general add-on to many other tasks. This could give broad impact. Being able to guarantee output consistency in a general way is a big bonus.

3. Extensive datapoints in experiment show a reliable observation on strong performances on exact path matching.

---

> ### Author Response · Authors · 2022-08-02
> **Response to Reviewer V2bR**
>
> We would like to thank the reviewer for their feedback, and are happy to see that they share our belief that SPLs promises to be an impactful addition to the community.
>
> Concerning the usage of different metrics for multi-label classification (MLC), we want to point out that scores such as hamming, jaccard and F1 scores, while very useful to provide an estimate of how many labels are correctly predicted, do not take into account at all if a complete label configuration is consistent with a constraint. This can be problematic in scenarios like ours where consistency matters. For these cases our recommendation is to not only report the usual MLC metrics, but also exact match and the number of constraints violations/satisfactions. Furthermore, it is important to inspect the predictions, as some configurations can not match the ground truth, but still be consistent and useful (as in our Warcraft experiment)

---

### Official Review · Reviewer_o1Rk · 2022-07-12

**Rating:** 7
**Confidence:** 3
**Soundness:** 3 good
**Presentation:** 4 excellent
**Contribution:** 4 excellent

**Summary:**

The paper introduces the semantic probabilistic layer (SPL), a replacement for the predictive layer of neural networks that is composed of probabilistic circuits and ensures the outputs satisfy given logical constraints. Then it is shown that the layer satisfies more desires than the competitors. Finally, SPLs are evaluated on simple path prediction, preference learning, shortest path finding, and hierarchical multi-label classification.

**Questions:**

Nothing to comment.

**Limitations:**

Nothing to report.

**Strengths And Weaknesses:**

The paper is really well written and the results are clearly presented. Despite that I don’t have a background on probabilistic circuits, I can still enjoy reading the paper.

The claims on the paper are well-supported through discussions and experimental sections. The authors compare SPL to the state-of-the-art methods through extensive experiments. So overall, I think the result is solid.

---

> ### Author Response · Authors · 2022-08-02
> **Response to Reviewer o1Rk**
>
> We would like to thank the reviewer for their encouraging words, and are glad they are excited about SPLs as much as we are.

---

### Meta-Review · Area_Chair_UZik · 2022-08-26

**Recommendation:** Accept
**Confidence:** Less certain

**Metareview:**

On the basis of the reviews I am recommending acceptance. A weakness for me is that neural models tend to learn logical constraints on structured labels directly from the data so in many cases enforcing the constraints does not improve performance. However, the idea of putting sum-product networks at the classification layer (probabilistic circuits) so that the partition function can be computed efficiently seems novel and interest enough to warrant publication.

**Award:**

No

---

### Decision · Program_Chairs · 2022-09-14

Accept